
# Tropospheric ozone maxima observed over the Arabian Sea during the pre-monsoon

**Jia Jia[1], Annette Ladstätter-Weißenmayer[1], Xuewei Hou[2], Alexei Rozanov[1] and John P. Burrows[1]**

[1]Institute of Environmental Physics, Bremen, Germany

[2]Nanjing University of Information Science and Technology, Nanjing, China

*Correspondence to*: Jia Jia (jia@iup.physik.uni-bremen.de)

**Abstract.** An enhancement of the tropospheric ozone column (TOC) over Arabian Sea (AS) during the pre-monsoon season is reported in this study. The potential sources of the AS spring ozone pool are investigated by use of multiple data sets (e.g., SCIAMACHY Limb-Nadir-Matching TOC, OMI/MLS TOC, TES TOC, MACC reanalysis data, MOZART-4 model and HYSPLIT model). 3/4 of the enhanced ozone concentrations are attributed to the 0-8 km height range. The main source of the ozone enhancement is considered to be caused by long range transport of ozone pollutants from India (~ 50% contributions to the lowest 4 km, ~ 20% contributions to the 4-8 km height range), the Middle East, Africa and Europe (~30% in total). In addition, the vertical pollution accumulation in the lower troposphere, especially at 4-8 km, was found to be important for the AS spring ozone pool formation. Local photochemistry, on the other hand, plays a negligible role in producing ozone at the 4-8 km height range. In the 0-4 km height range, ozone is quickly removed by wet-deposition. The AS spring TOC maxima are influenced by the dynamical variations caused by the sea surface temperature (SST) anomaly during the El Niño period in 2005 and 2010 with a ~5 DU decrease.





## 1 Introduction

Tropospheric ozone is one of the most important green-house gases and one of the most important components of photochemical smog. Most tropospheric ozone is produced in situ by photochemical reactions of its precursors ($NO_x$ ($NO + NO_2$), CO, $CH_4$ and VOCs) in the presence of sunlight, while some tropospheric ozone naturally originates in the stratosphere. High surface ozone values are detrimental to human health by causing respiratory illnesses, and can also lead to losses in agricultural crops (see Van Dingenen et al., 2009; Mills et al., 2016 and references therein).

In this study, we investigated the global pattern of tropospheric ozone by averaging 7 years (2005-2011) of Tropospheric Ozone Column (TOC) data products from different satellite instrumentation: SCIAMACHY Limb-Nadir-Matching TOC (Ebojie et al., 2014; Jia, 2016) and the OMI/MLS TOC (Ziemke et al., 2006). A tropospheric ozone maximum is observed over the Arabian Sea (AS). This enhancement of TOC can be observed in yearly mean image as well (Fig. 1). The enhancement of TOC is similar in magnitude as TOC enhancements observed during the follow events: 1) the well-known biomass burning plume in the Southern Hemisphere that was transported over the South Atlantic, the coast of South Africa, along the Indian Ocean and towards Australia (e.g., Fishman et al., 1986, 1991; Pickering et al, 1996; Thompson et al., 1996, 2001; Lelieveld and Dentener, 2000; Staudt et al., 2002; Sinha et al., 2004), 2) TOC attributed to anthropogenic sources in the Northern Hemisphere, and 3) the Mediterranean summer ozone pool attributed to the stratospheric-tropospheric exchange (STE) (Zanis et al., 2014). A spring (or so called pre-monsoon, see Sect. 2.1) TOC maximum of ~42 DU on monthly average was identified from our study of the seasonality of the TOC.

Spring maxima in TOC are not unique over the AS but rather a well-known large scale phenomenon in the Northern Hemisphere. Nevertheless, the origin and mechanisms explaining this phenomenon is still a matter for debate (e.g. Monks, 2000, 2015 and references therein). The increase of tropospheric pollutants, presumably increase of longer lived VOCs which are ozone precursors, during winter, may play an important role by influencing the two major contributors to tropospheric ozone concentrations: the STE intrusions and the photochemical production process (Holton et al.,1995; Penkett et al., 1998; Monks, 2000). In a remote region like AS, an intuitive hypothesis is that long range transport (LRT) of ozone from more polluted regions or from STE may be the drivers. This is because of the longer ozone lifetime in spring and the weak local production over remote areas (Wang et al., 1998).

In previous studies using the ozonesonde measurements above the west coast of India and the data from two campaigns (the 1998 and 1999 INDOEX – INDian Ocean EXperiment campaigns and the ICARB – Integrated Campaign for Aerosols, Gases and Radiation Budget campaign which was conducted during March-May 2006), the higher AS TOC during pre-monsoon season was confirmed to be significantly influenced by LRT of the continental anthropogenically influenced outflows from the Middle East, Western India, Africa, North America and Europe (Lal and Lawrence, 2001; Chand et al, 2003; Srivastava et al., 2011, 2012; Lal et al., 2013, 2014). In addition, by comparing the INDOEX ozone measurements from both sides (northern and southern) of the ITCZ (InterTropical Convergence Zone), the influence of the ITCZ functioning as a sink for ozone was determined by the observed 4 times higher TOC values on the northern side of AS compared to the southern side (Chand et al., 2003). The seasonal variation of tropospheric ozone at Ahmadabad (23.03 °N, 72.54 °E) was reported to have an



averaged maximum of ~44 DU in April during the years 2003–2007 (Lal et al., 2014). The possibility of the
STE influencing the ozone mixing ratio up to ~10 km altitude was also discussed. However, the mechanisms
explaining this phenomenon need to be better understood.



Figure 1. Yearly average for TOC retrieved from (left) SCIAMACHY LNM and (right) OMI/MLS in 2008.
Here, the TOC enhancement over the AS is investigated and interpreted by using TOC data products from sev-
eral satellite remote sensors (i.e. SCIAMACHY Limb-Nadir Matching, OMI/MLS and TES), MACC (Monitor-
ing Atmospheric Composition and Climate) reanalysis data (Inness et al., 2013) and simulations from the global
tropospheric chemical transport model (CTM) MOZART-4 model (Model for Ozone and Related Tracers)
(Emmons et al., 2010). This study focuses on the analysis of the regional contribution to LRT, the influence of
the meteorological conditions, the local chemistry and STE, and the inter-annual variability of the spring ozone
maxima, thus to better understand the climate interact with the distribution of tropospheric ozone through tem-
perature, humidity and dynamics. In Sect. 2, the data sets used in this study are briefly discussed. In Sect. 3, the
regional distribution and the time series of tropospheric ozone and its precursors are investigated. Meteorologi-
cal and photochemical sources of ozone plumes due to LRT, local chemistry and STE are discussed in Sect. 4.
The role of accumulation of pollutants is also highlighted in this section. In Sect. 5 the impact of El Niño on the
inter-annual variability is identified. Finally, conclusions are given in Sect. 6.
**2    Data sets used in this study**
The SCanning Imaging Absorption spectroMeter for Atmospheric CHartographY (SCIAMACHY) was a pas-
sive spectrometer designed to measure radiances in eight spectral channels, covering a wide range from 214 nm
to 2384 nm with a moderate spectral resolution of 0.21 nm to 1.56 nm (Burrows et al., 1995; Bovensmann et al.,
1999). SCIAMACHY performed observations in three viewing modes: nadir, limb and solar/lunar occultation.
The SCIAMACHY Limb-Nadir-Matching TOC is retrieved based on the tropospheric ozone residual (TOR)
method, which subtracting the stratospheric ozone columns retrieved from the limb measurements, from the
collocated total ozone columns acquired from nadir measurements, by using the tropopause height data (Ebojie



et al., 2014). The results showed in this study are from the V1.2 SCIAMACHY Limb-Nadir-Matching TOC data
set. This data set is recently developed in the Institute of Environmental Physics (IUP) in the University of Bre-
men (Details can be found in Jia, 2016). The data set is not complete in a full SCIAMACHY performing time
period (2002-2012), thus is not showed in the time series image.
The UV-Vis nadir viewing spectrometer Ozone Monitoring Instrument (OMI), the thermal-emission Microwave
Limb Sounder (MLS) and the infrared Fourier transform spectrometer Tropospheric Emission Spectrometer
(TES) are three of the main instruments onboard the EOS Aura satellite (Levelt et al., 2006; Waters et al., 2006;
Beer, 2006). The OMI/MLS TOC data set is retrieved based on TOR method using data sets from OMI and
MLS. The adjustment for inter-calibration differences of OMI and MLS instruments is performed by using the
CCD method (Ziemke et al., 2006; 2011). Both OMI-CCD and MLS measurements of the stratospheric ozone
are averaged for the comparison over the Pacific ($120\,°W$-$120\,°E$) (Ziemke et al., 2006). The MLS data is ad-
justed according to the observed differences, and then interpolated in two steps along-track and along longitude.
In the end, OMI/MLS is able to provide daily-based global TOCs.
TES ozone is retrieved from the 9.6 μm ozone absorption band using the $995 - 1070$ cm$^{-1}$ spectral range. In
cloud-free conditions, the nadir vertical profiles have around four degrees of freedom (DOF) for signal, ap-
proximately two of which are in the troposphere, giving an estimated vertical resolution of about 6 km with a
footprint of $5.3\,km \times 8.5\,km$, covering an altitude range of 0-33 km (see Beer et al., 2001; Nassar et al., 2008
and the references therein).
MACC is a research project for the European GMES (Global Monitoring for Environment and Security) initia-
tive (Inness et al., 2013). MACC combines a wealth of atmospheric composition data with a state-of-the-art nu-
merical model and data assimilation system to produce a reanalysis of the atmospheric composition. MACC
reanalysis data of ozone, CO and specific humidity used in this study are available in 6-h time intervals (00, 06,
12 and 18 UTC) and were provided in monthly files with the unit of kg/kg under the website
http://apps.ecmwf.int/datasets/data/macc-reanalysis/levtype=ml/. The horizontal resolution of the model is $1.125\,°$
$\times 1.125\,°$. Variables were provided as 3D fields in pressure hybrid vertical coordinates. The vertical coordinate
system is given by 60 hybrid sigma-pressure levels, with a model top at 0.1 hPa.
MOZART-4 is a global tropospheric CTM. It was run with the standard chemical mechanism (see Emmons et
al., 2010 for details) in this study. MOZART-4 was driven by the NCEP/ National Center for Atmospheric Re-
search (NCAR) reanalysis meteorological parameters, having a horizontal resolution of approximately $2.8\,° \times$
$2.8\,°$, with 28 vertical levels from the surface to approximately 2 hPa. The chemical initial condition in 2000 and
emissions from 1997 to 2007 used in MOZART-4 were from the NCAR Community Data Portal
(http://cdp.ucar.edu/), which was introduced by Emmons et al. (2010). The model was run with a time step of 20
min from 1 January 1996 to 31 December 2007, and the first year was discarded as spin-up (Hou et al., 2014;
Zhu et al., 2015). The tagged tracer method was used to isolate the contributions from individual source regions.
This method was introduced by Sudo and Akimoto (2007). It treats a chemical species emitted or chemically
produced in a certain region as a separate tracer and calculates its transport, chemical transformation and surface
deposition. The results used in this study are simulated by Hou et al. (2014).



HYbrid Single-Particle Lagrangian Integrated Trajectory (HYSPLIT) is a system for the computation of simple
air parcel trajectories from the National Oceanic and Atmospheric Administration (NOAA). In order to investi-
gate the forward and backward trajectory of the air mass, the web-based version of the HYSPLIT model (Stein
et al., 2015) is used for this study: http://ready.arl.noaa.gov/hypub-bin/trajtype.pl?runtype=archive.
**3    Observation of a pre-monsoon enhancement in TOC data products**
Satellite retrieved TOCs have a better spatial and temporal coverage compared to ozonesonde measurements.
However in situ measurements of ozone from ozonesondes are considered more accurate. Combining the two
types of measurements provides an opportunity to investigate data-sparse regions such as the AS. In comparison
with the studies using ozonesonde and ship measurements, the analysis of the satellite observations of TOCs
regarding AS region is still a gap in the current state. In this section, the satellite observations and the model
results are presented.
Figure 2 shows the regional distribution of the TOC and two of its photochemical precursors: $NO_2$ and CO. Sea-
sonal cycles of TOC, CO and $NO_2$ over the AS are shown in Fig. 3. A seasonal pattern of TOC is observed in
both OMI/MLS and TES. An offset of ~5 DU exists between the two investigated TOC data products. A maxi-
mum of TOC over AS is observed (~42/47 DU) in every April during the years 2005–2012, followed by mon-
soon/summer minima of ~20 DU. TOC recovers to ~35 DU in the post monsoon autumn but drops down
slightly during the winter monsoon. This seasonal pattern is consistent with the results from the sonde station
Ahmedabad (Fig. 6 in Lal et al., 2014) and depends on the meteorological conditions (Sect. 3.1). The ozone pre-
cursors, CO and $NO_2$, show a different behaviour than ozone. As $NO_2$ has a short lifetime (2-8 hours, Beirle et
al., 2011), the tropospheric $NO_2$ data products, retrieved from observations of SCIAMACHY or other related
instrumentation in space, show high values over anthropogenic sources and relatively low values, often below
the detection limit, over the remote regions. Over the AS, tropospheric $NO_2$ columns are small being around
$10^{14}$ molec/cm$^2$. This small concentration originates from ship emissions and continental outflow (Richter et al.,
2004). Higher values can be observed during the winter monsoon from transport off the Asian coast. CO has a
longer life time than ozone (~2 months in average for CO and ~23 days for tropospheric ozone, Novelli et al.,
1998; Young et al., 2013). Due to the relatively long lifetime of CO and ozone, both trace gases show a similar
transport pattern. For instance, the biomass burning plume originating from Southern Africa in boreal autumn in
the Southern Hemisphere can be observed as well for CO as for ozone. However, in comparison to ozone, which
is produced due to the photochemical production, the spatial pattern of CO is known to be more driven by emis-
sions than dynamical processes (Logan et al., 2008). Thus, the time series of the data products for tropospheric
CO reveal a similar winter maximum as $NO_2$, and it also shows a smaller peak in spring time as TOC. The
spring peaks of CO are observed one month earlier than that those of TOC. This shift can be caused by the com-
bustion emission of CO in Southern Asia (Fig. 6 in Duncan et al., 2003).







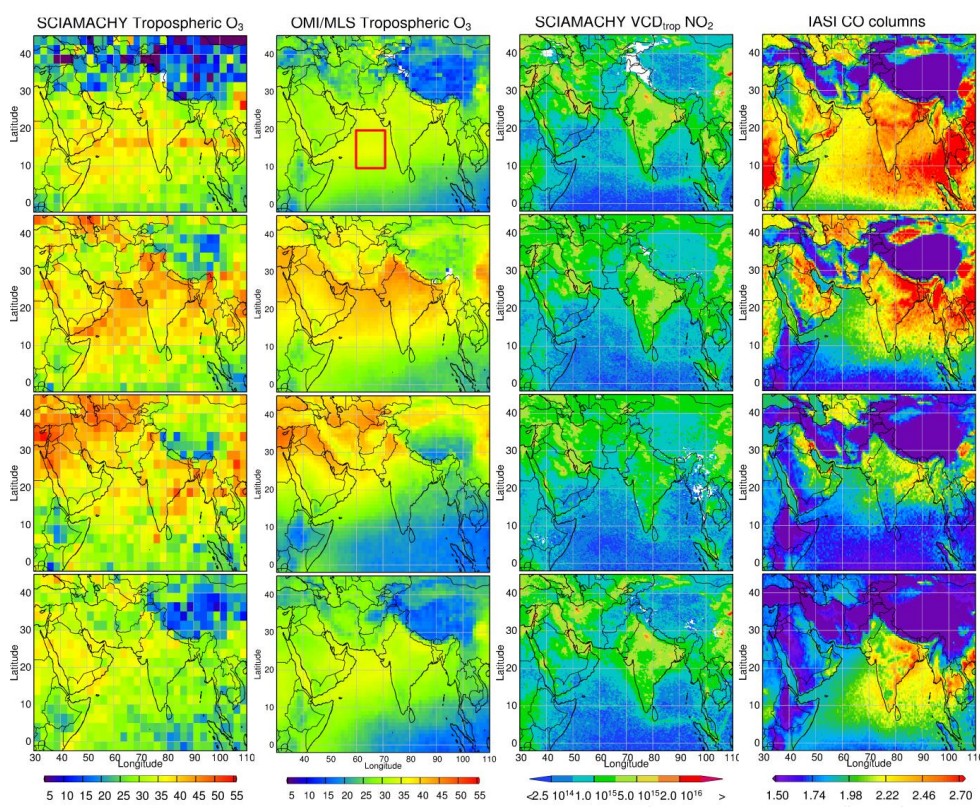

Figure 2. Plots of the TOC, $NO_2$ (Hilboll et al., 2013) and CO ($\times 10^{18}$, George et al., 2009) as a function of season in 2008. The unit for TOC are DU and that for $NO_2$ and CO in molec/$cm^2$. From top to bottom are DJF (December to February), MAM (March to May), JJA (June to August) and SON (September to November).

In this manuscript, MACC reanalysis data is used to provide vertical information of ozone. This choice is motivated by the fact that OMI and MLS satellite ozone data were actively assimilated in the MACC reanalysis and constrain tropospheric ozone (Inness et al., 2013). Figure 4 shows MACC results of ozone partial columns between 0 km and the TPH, determined from ECMWF retrieval (Ebojie et al., 2014), and between 0-8 km, respectively. In a year with no ENSO (El Nino-Southern Oscillation) event in spring, for instance in 2006 (upper panels of Fig. 4), the enhanced ozone during pre-monsoon is ~30 DU out of ~40 DU (~3/4) originating from the lower troposphere (0-8 km). Because of this result, possible origins will be discussed in the following section (Sect. 3) by analysing 4 various altitude ranges: 0-4 km, 4-8 km, 8-12 km and 12-18 km.



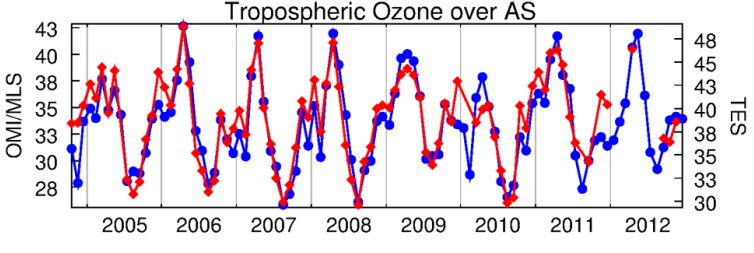

Figure 3. Trace gas time series over AS (10-20 °N, 60-70 °E) from 2004 to 2012. The blue (solid and dotted)
curves represents OMI/MLS ozone, red is TES ozone, magenta is IASI CO and black stands for SCIAMACHY
$NO_2$. The vertical columns are given in DU for ozone and molec/$cm^2$ for $NO_2$ and CO. The region used for this
time series calculation is marked with red rectangle in Fig. 2. The time series of TOC from SCIAMACHY,
MACC, OMI.MLS and TES in the year 2008 is presented in appendix Fig. A3.
**4    Potential origins of the AS pre-monsoon ozone pool**
**4.1    Influences of meteorology**
The AS region is defined in this study as 10-20 °N, 60-70 °E on the west side of the sub-continental India. This
location is influenced by the tropical/subtropical air mass exchanges and the sea breeze circulation (e.g., Law-
rence and Lelieveld, 2010). The climate of AS can be divided into 4 different seasons, due to the seasonal varia-
tion of the ITCZ: winter-spring monsoon (Dec-Feb), pre-monsoon transition (Mar-May), summer monsoon
(Jun-Aug), and post-monsoon transition (Sep-Nov). In summer, the ITCZ is at its northernmost position. The
wind appears to be westerly and strong due to the Somali jet (Fig. 5). This condition causes strong precipitation,
higher cloud cover frequency and increased air humidity over AS (David and Nair, 2013). The wind and strong
precipitation 'wash' the air masses and remove soluble pollutants. A summer monsoon minimum for the trace
gases such as shown in Fig. 3 may be expected. The destruction of ozone by reactive halogens is another poten-
tial sink for the ozone in the marine boundary layer (Dickerson et al., 1999; Ali et al., 2009). During the pre-
monsoon transition, surface winds are westerly at the northern AS with an anticyclonic pattern centred over the
middle AS. At this time, the AS is most of the time cloud-free and dynamically steady. This cloud-free anti-
cyclonic condition possibly causes subsidence of air masses and results in accumulation of pollutants (Sect. 3.2).



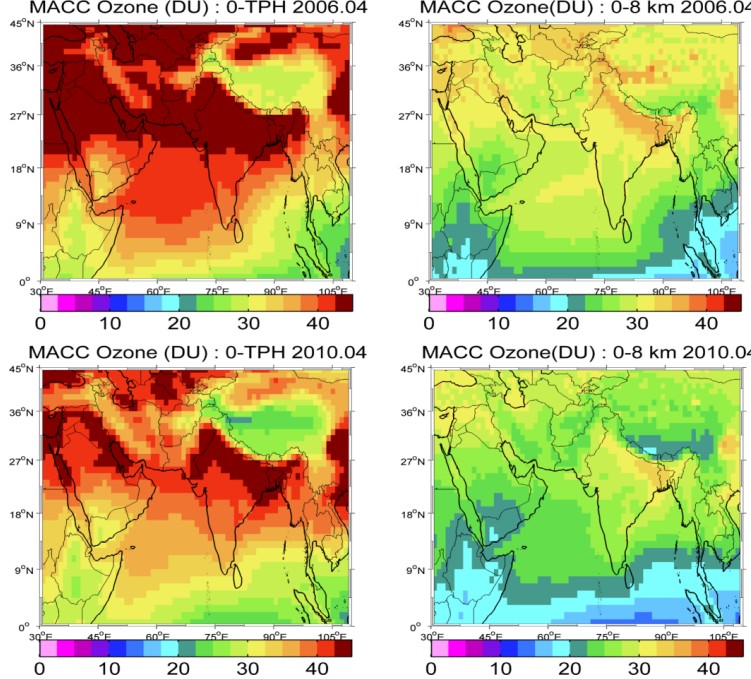

Figure 4. Ozone partial columns (TOC in the left panels and 0-8 km column of tropospheric ozone in the right
panels) from MACC reanalysis model in April 2006 (upper panels) and 2010 (lower panels).
The ITCZ located at the southern part of the AS can become the 'border' that stops the pollutants (in this case,
tropospheric ozone) diffusing to the Indian ocean with ozone depletion on the surface of cloud droplets in the
convective region (Lelieveld and Crutzen, 1990).
It is worth mentioning that the solar radiation over the AS, unlike in the middle/high latitudes where it is strong-
est in summer, reaches its maximum during pre-monsoon (Weller et al., 1998; David and Nair, 2013). One
could argue that a maximum solar radiation can cause stronger photochemical reactions and thus an increased
ozone concentration. To answer this question, the contribution of the photochemical production will be investi-
gated in Sect. 3.3.



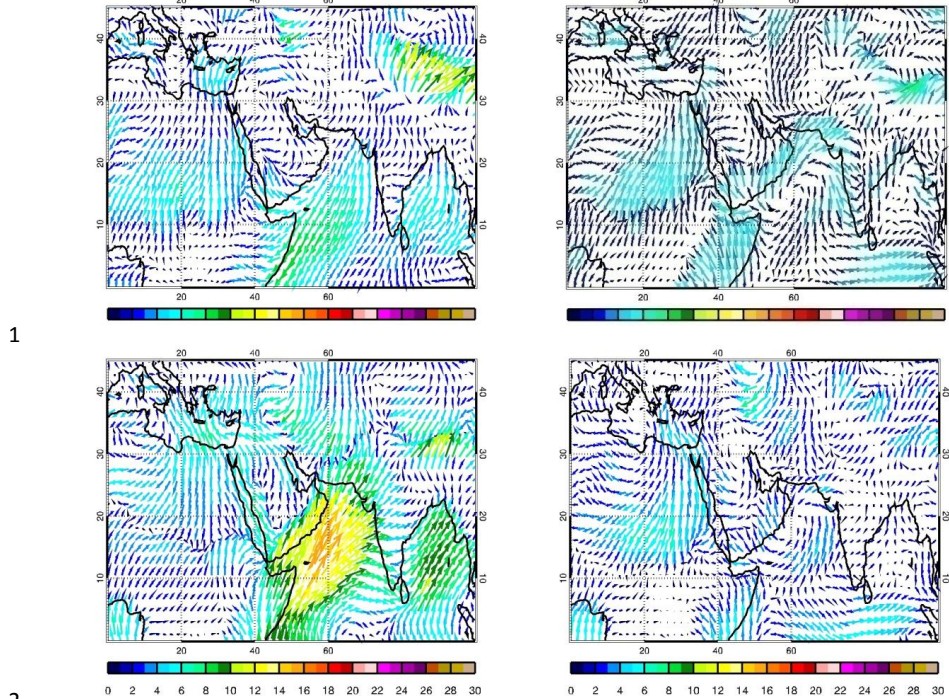

Figure 5. 10 meter sea surface wind on January (upper left), April (upper right), July (lower left) and October (lower right) over AS at 2008 from NCEP (National Center Environmental Prediction). Figure provided by Anne Blechschmidt from the University of Bremen.

## 4.2 Long range transport mechanism and pollutant accumulation

It is established in the previous studies that LRT plays an important role in the AS pre-monsoon ozone pool (Lal and Lawrence, 2001; Chand et al, 2003; Srivastava et al., 2011, 2012; Lal et al., 2013, 2014). For example, the satellite data products for CO and TOC are highly correlated (Fig. 3). Trajectory models are used to investigate the LRT pathways of the air parcels. Figures 6 and 7 show an example of the HYSPLIT backward and forward trajectory results for air masses over AS at 2, 4, 7 and 15 km in April 2008. In the lower troposphere (0-8 km), the sources are identified as the Middle East, India and North Africa, which are consistent with the previous studies. The higher tropospheric ozone (12-18 km) is found in air which was uplifted and transported from the North Indian Ocean and Southeast Asia. The air masses in the lower troposphere subside 4-5 km locally within a high pressure system within 10 days. This confirms the conclusion on accumulation of pollutant which was derived from the wind field information in Sect. 3.1 (see Fig. 5). This theory was also proved by Srivastava et al. (2011) from the TPSCF (Total Potential Source Contribution Function) results. One explanation for the larger TOC over the AS in comparison to surrounded regions is the lower humidity which provides less favorable con-





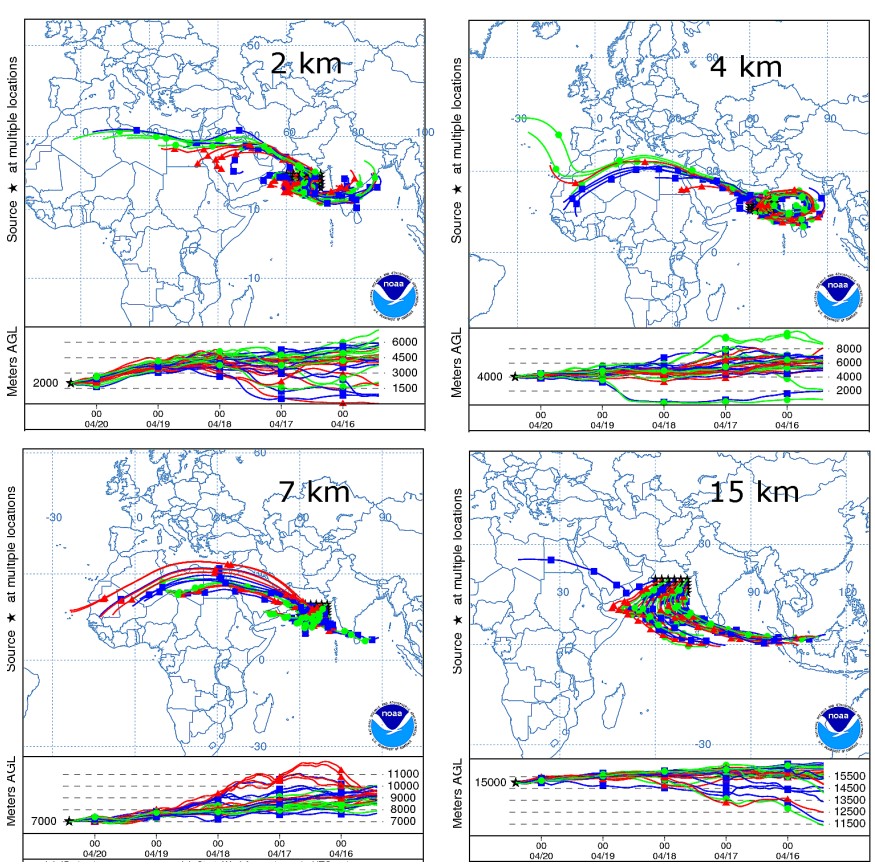

Figure 6. HYSPLIT trajectory 120 hr backward model results for air masses at AS with source location at 2, 4, 7
and 15 km.
dition for ozone depletion by Hydroxyl radicals (OH) (Fig. 12). This is further discussed in Sect. 3.3. In addition
to the sources, here we also investigate the areas that are influenced by the AS ozone pool (Fig. 7). The ozone-
rich air over the AS is transported back to India (lower left panel). HYSPLIT also simulates transport to the Red
Sea through the Gulf of Aden in the lower troposphere (upper panels), which is expected because of the moun-
tains aside acting as a barrier for pollution transport. The elevated tropospheric air masses are also transported
towards the Pacific Ocean via China (lower right panel).
To quantify contributions to LRT from different source locations, the tagged tracer simulation with MOZART-4





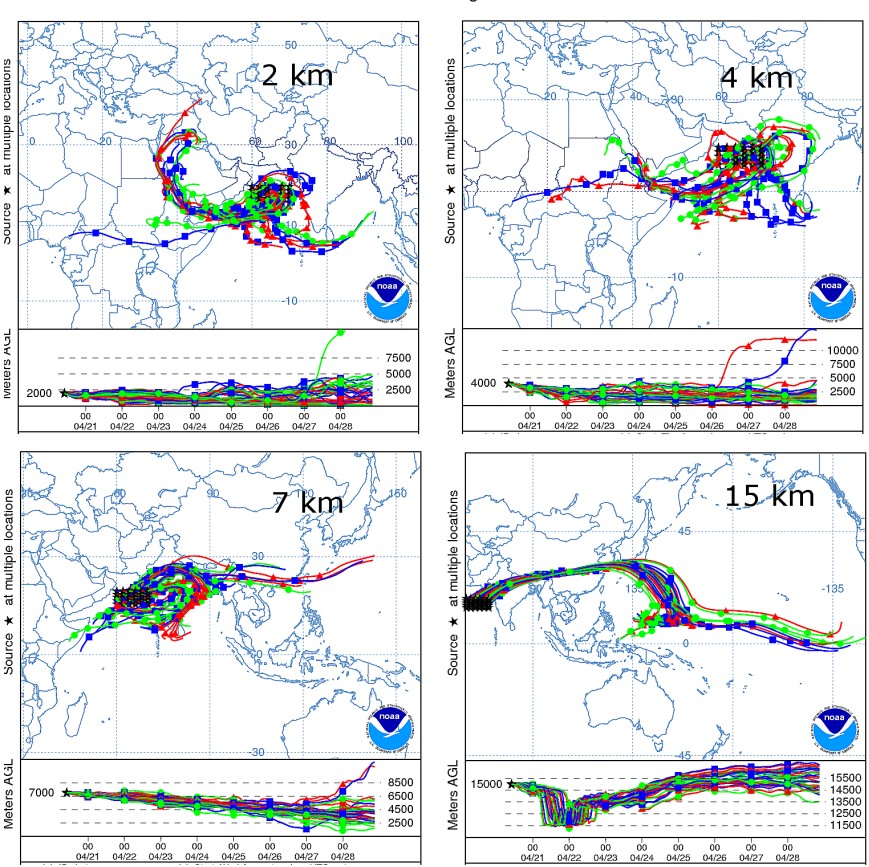

Figure 7. HYSPLIT trajectory 240 hr forward model results for air masses at AS with source location at 2, 4, 7 and 15 km.

CTM (Sudo and Akimoto, 2007; Hou et al., 2014) during 1997-2007 was used. Figure 8 shows the seven tagged regions. Europe, Africa and the Middle East are combined into one hot spot as the closer western region (named 'Euro'). India, Bay of Bengal and AS are presented together as the closer eastern region (named 'India'). Note that when evaluating the contribution from this region, the influence from pollutant accumulation over the AS should always be considered. The Indian Ocean is included in the 'Rest' region. 'NA' and 'SA' represent North America and South America respectively. The regions are divided due to the time consumption of the model calculation. For further studies, another arrangement can be considered.



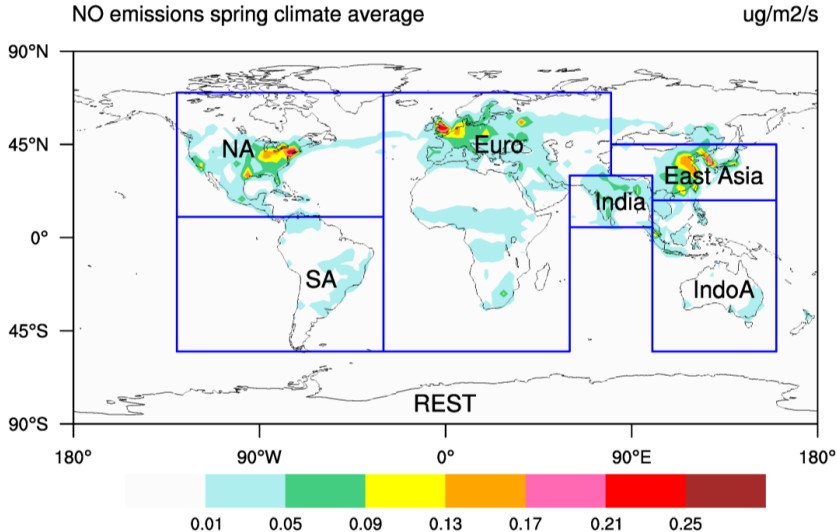

Figure 8. Regional separation for tracer tagging with distributions of the spring mean emission rate ($\mu g/m^2/s$) of
NO (including anthropogenic, biomass burning, and soil emissions) at the surface used in the model simulations
during 1997–2007.
The source region distribution varies for different altitude ranges (Fig. 9). In the 0-4 km layer, ~30% of the
transported ozone comes from the 'Euro' region. The 'India' region is the biggest source region that contributes
50%, of which 60% comes from the boundary layer. In the 4-8 km layer, the influences of the boundary layers
are much smaller, while 'Euro_FT' contributes ~10% more than to the 0-4 km layer. The far-away source re-
gions ('NA', 'SA' and 'IndoA') become non-negligible (~10% each). The contribution from 'IndoA' increases
with height. Since the Indian Ocean is included, an increased contribution from 'Rest' with altitude is expected.
In conclusion, the main contributor to LRT is 'Euro_FT' with 30% contribution in average, followed by the 'In-
dia' region with over 20% contribution. The inputs from far-away source regions are similar, with ~10% each.
The influence from East Asia is negligible. Note that the air masses in the higher altitudes are normally quickly
removed by the strong advection (Fig. 10). The contributors in the lower altitudes (<12 km) have more influ-
ences on the ozone accumulation.



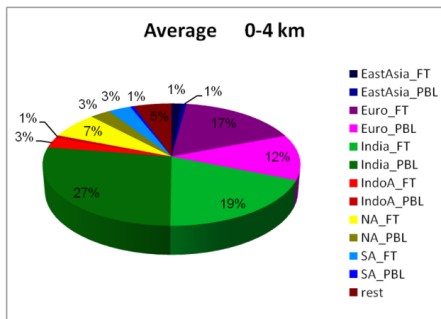
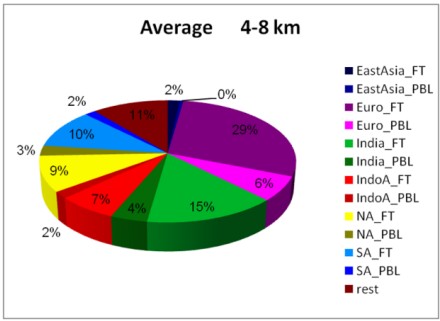

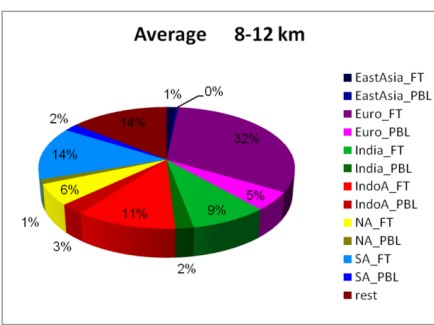
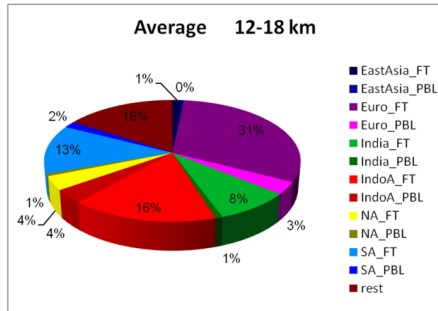

Figure 9. Averaged LRT contributions to the AS tropospheric ozone concentration from different source regions
to 4 atmospheric layers over the AS in April 1997-2007. PBL (planetary boundary layer) is defined as the region
from surface to the top of the boundary layer. FT (free troposphere) is defined as extending to the tropopause
above the BL.
**4.3    Local chemistry**
This section addresses two questions: (1), What is the role of the photochemistry for TOC above AS? (2), Is
more ozone been photochemically produced during the long accumulation time in the middle (4-8 km) or lower
(0-4 km) troposphere?
The ozone budget is calculated in the MOZART-4 model (Fig. 6.11) within the 1997-2007 time periods. Photo-
chemistry plays a very different role in the four altitude ranges. In the 0-4 km layer, water vapour acts as a
source of OH radicals and depletes ozone. Compared to the photochemical production, this depletion process
dominates (Nair et al., 2011). Thus a net outflow of ozone in chemistry was observed. In the higher layers (8-12
km, 12-18 km), photochemical production becomes a major source of ozone, while advection being the major





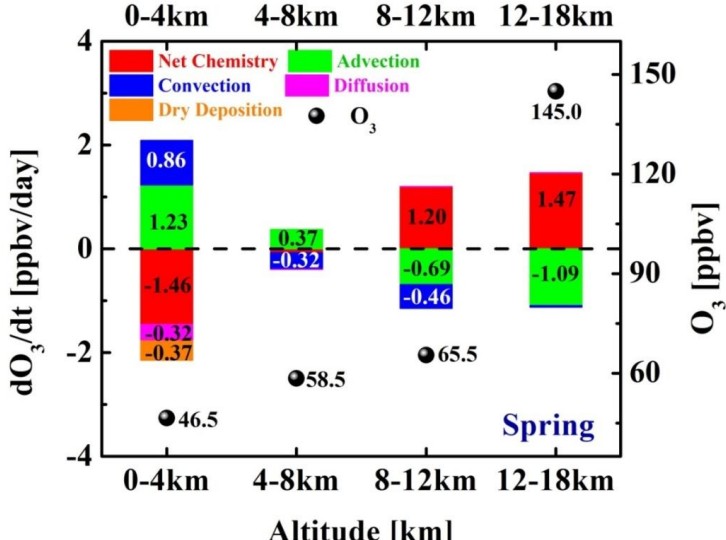

Figure 10. Averaged ozone budget in pre-monsoon from MOZART-4 at four layers over AS region. The ozone
partial column volumes (ppbv) calculated from MOZART-4 is presented as black dots.
sink. Zahn et al. (2002) estimated that the annual net ozone production rates over AS are $17.6 \times 10^{10}$ molecules
$cm^{-2} s^{-1}$, by using the CARIBIC (Civil Aircraft for the Regular Investigation of the atmosphere Based on an
Instrumented Container) aircraft data from 10-11 km altitude. However, Livesey et al. (2013) showed in MLS
data that such maximum of ozone amount around 215 hPa (~ 11 km) in pre-monsoon season is most likely a
zonal pattern. In the 4-8 km layer, the budget is rather small with a net inflow by advection. The net chemistry is
less than -0.1 ppbv per day, indicating a negligible sink.
The $O_3$-CO correlation has been broadly used to indicate tropospheric $O_3$ sources (Fishman and Seiler, 1983;
Kim et al., 2013; Inness et al., 2015). A positive $O_3$-CO correlation denotes considerable chemical production of
$O_3$. A negative correlation, on the other hand, can originate from chemical $O_3$ loss or deposition, or can suggest
that the air mass is either transported from the stratosphere, or moved by advection from the free troposphere.
Figure 11 shows the $O_3$-CO correlation at 4-8 km in April using MACC data. The left panel is a typical correla-
tion result with strong negative correlations over AS (See also appendix Fig. A1). This correlation suggests that



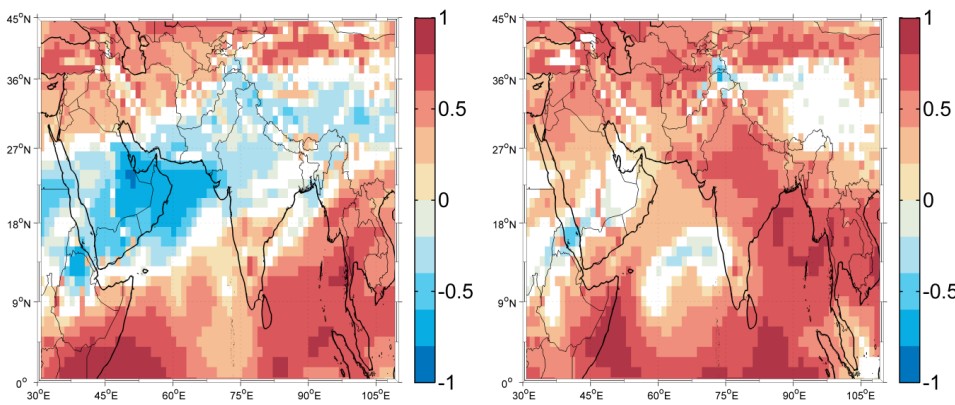

Figure 11. $O_3$-CO correlations calculated for 4-8 km column abundances with 3 hr temporal interval in April

2008 (left panel) and 2006 (right panel) from MACC reanalysis data.

a chemical production of ozone is most likely not the cause for the observed ozone enhancement. Some ozone

production is expected (e.g., year 2006 in the right panel), but this kind of situation is rare.

The averaged specific humidity in-between 4-8 km was used to investigate evidence for the impact of $HO_x$ re-

moval on ozone in clean air conditions (Fig. 12):

$$OH + CO + O_2 \rightarrow HO_2 + CO_2$$

$$HO_2 + O_3 \rightarrow OH + 2O_2$$

Net: $CO + O_3 \rightarrow CO_2 + O_2$

Unlike the humid lowest troposphere, the air masses over the AS at 4-8 km are rather dry compared to the sur-

roundings. This can be explained by adiabatic lifting and expansion of marine boundary air followed by conden-

sation and removal for $H_2O$. The lifting is stronger over land than over ocean due to the temperature differences.

This is also one of the reasons that Southeast Asia has strongest convection. The dry air at 4-8 km can lead to a

smaller depletion contribution from OH radicals, thus it is more suitable for ozone accumulation. Hence, the AS

ozone columns in the 4-8 km altitude region are expected to be higher than its surroundings.





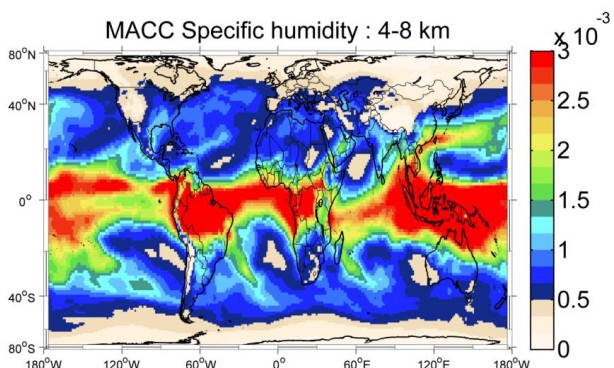

Figure 12.  Specific humidity (kg/kg) at 4-8 km in April 2006 from MACC reanalysis dataset.
**4.4    Stratosphere-troposphere exchange**
The STE is not the focus of this study, thus is only briefly mentioned here. The ozone concentrations in the ex-
tra-tropical lower stratosphere show a maximum in late winter/early spring as driven by the Brewer-Dobson
circulation (e.g., Fortuin and Kelder, 1998; IPCC/TEAP, 2005). Fadnavis et al (2010) indicated ozone strato-
spheric intrusion during winter and pre-monsoon season over the Indian region (5-40 °N, 65-100 °E) by using
both satellite and model data. One stratospheric intrusion was observed over AS during the ICARB campaign
(Lal et al., 2013) in 5 May 2006 (Fig. 13). However, it is not clear yet how much and how deep the influence
can be. In our study, the STE contribution simulated by MOZART-4 tagged tracer method is comparable with
the ones transported from 'Euro_FT' in each altitude range. The STE origin might be a reason for the strong
negative $O_3$-CO correlation since the chemical loss and deposition are excluded (Sect. 3.3).

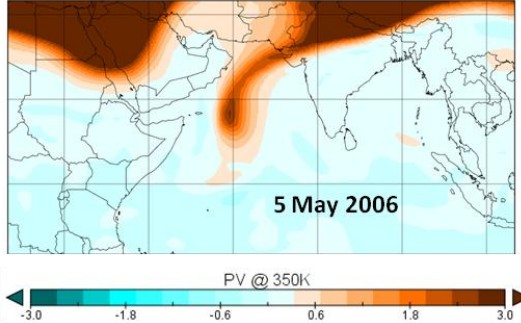

Figure 13. Potential Vorticity from ECMWF.



## 5    ENSO and Interannual variation

Two spring anomalies are depicted in 2005 and 2010 where ozone is ~5 DU lower compared to other years (upper panel of Fig. 3). The decrease in 2010 is most likely to be the anomaly of the lower troposphere ozone as observed in Fig. 4. The two following facts suggest the anomaly to be dynamical: (1), the ozone reduced in a similar amount at continental surroundings; (2), similar to ozone, a lower CO maximum appeared in 2010 (lower panel of Fig. 3).

The El Niño events, as driven by a reversal of the Walker Circulation, affect the temperature, humidity and biomass burning emissions, thus influence the trace gases including ozone. Particularly, the tropospheric ozone anomaly related to the El Niño event during the year 1997-1998 was intensively studied (e.g. Chandra et al., 1998; Sudo and Takahashi, 2001). The tropospheric ozone increased (up to 25 DU in the burning season) over the equatorial western Pacific due to a reduced convection and growing burning emissions, whereas decreased (4-8 DU) over the eastern Pacific because of the change in meteorological conditions. By using a model simulation, Zeng and Pyle (2005) reported that the tropospheric ozone concentration at specifically the equatorial region 40-70 °E decreases with similar amount as over the eastern Pacific during El Niño events. Ziemke et al. (2010) showed that the ENSO related response of tropospheric ozone over the western and eastern Pacific dominated interannual variability. An Ozone ENSO Index (OEI) was formed to represent the ENSO impact. The OEI was calculated by subtracting the eastern and central tropical Pacific region tropospheric ozone (15 °S-15 °N, 110-180 °W) from the western tropical Pacific-Indian Ocean region (15 °S-15 °N, 70-140 °E) with the fact that the zonal variability of tropic stratospheric ozone is only ~1 DU.

Figure 14 shows the OEI index that is produced from OMI/MLS data for the related time period (dark red curve). A 'correction' of OMI/MLS TOCs over the AS by adding the OEI index is performed. The ozone spring maxima anomalies at pre-monsoon season in 2005 and 2010 (blue curve) can no more be seen after the 'correction'. This indicates that the El Niño induced dynamics might contribute to the interannual variability over pre-monsoon AS ozone. Since OEI contains both chemical (fire) and dynamical influences in the burning season, ozone peaks (in the red curve) can be observed in the winter of 2006 and 2009 when strong fires happened in Indonesia.

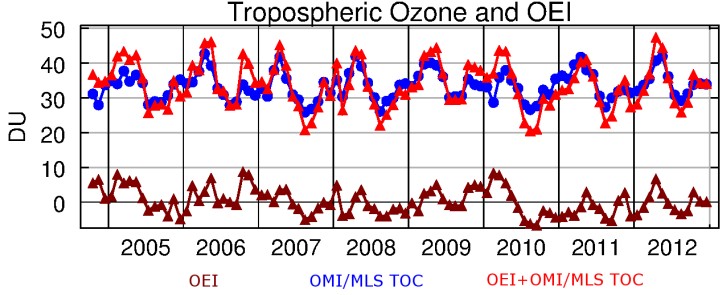

Figure 14. Time series of tropospheric ozone columns 'corrected' with OEI over AS (10-20 °N, 60-70 °E) from 2004 to 2012. The blue curve represents OMI/MLS TOCs, dark red is OEI calculated from OMI/MLS and red stands for OMI/MLS TOCs with OEI 'correction'.



The dynamical influence of El Niño can be found in two aspects. El Niño can induce an increase of Sea Surface
Temperature (SST) thus strengthens the water vapor upwelling to the middle troposphere, and then reduces the
life time of ozone. It also possibly triggers changes in STE flux as mentioned by e.g. Neu et al. (2014). Moister
air masses are observed from MACC reanalysis data in April 2010 (appendix Fig. A2). This confirms the as-
sumption of the SST influence over AS. A STE flux variation can be caused by both El Niño and La Niña
events. In this case La Niña events (in 2011) didn't contribute as much as El Niño events (in 2005 and 2010).
The impact of El Niño is mainly expressed by the SST anomaly instead of the STE anomaly.

## 10  6    Conclusions

The 7 years composite averaged values for TOC presented over AS exhibit a seasonal pattern and have values
similar to those in the Southern hemispheric biomass burning plume. A disciplined tropospheric ozone seasonal-
ity with a ~42 DU maximum at the pre-monsoon season was shown in the satellite based OMI/MLS and TES
observations as well as in the MACC reanalysis model. The seasonal feature is found to be strongly related to
the meteorological conditions.
Previous studies illustrated the importance of LRT to the pre-monsoon ozone enhancement and confirmed the
source locations to be the Middle East, West India, Africa, North America and Europe. Here various regional
contributions to the AS pre-monsoon ozone through LRT were analysed by dividing the global range into 7 re-
gions using the MOZART-4 tagging tracer simulation method. In the lowest 4 km, the sources from India con-
tributed ~50% of the transported AS ozone amount. The free troposphere of the Middle East, Africa and Europe
(so called 'Euro_FT') started to play a major role from 4 km altitude and higher. The contribution is on average
30%. The Indian region is still the second important source region at 4-8 km with ~20%. Its contribution is
slowly replaced by the further-away source regions at higher altitude range. It is worth mentioning that South
America plays a more important role compared to North America, yet there is no explanation for this result so
far.
In addition, the vertical pollutant accumulation in the lower troposphere, especially at 4-8 km, is important to
the AS spring ozone pool. The suitable meteorological conditions were discovered from wind field data from
NECP and specific humidity data from MACC. First, the cloud-free anticyclonic condition that is observed from
the wind field data can cause air to be transported upside down. This point is supported by the forward model
results of HYSPLIT showing that at ~7/8 km or lower, the air circles down over the AS region for around 10
days without diffusion. Second, at 4-8 km the air over AS is much dryer than the surroundings. This is most
probably due to relatively lower temperature over the sea which caused that the moisture cannot be lifted up as
high as over land. The dry conditions induce the accumulation of ozone with a longer life time thus cause the
AS ozone to be outstanding from the sub continental regions.
The averaged spring ozone budget was calculated using MOZART-4 to improve our understanding of the addi-
tional local chemical activity. Ozone is photochemically produced at high altitudes (8-18 km) and is removed by
advection. In the lowest 4 km ozone is depleted by OH radicals. Positive ozone budgets from advection and
convection can be observed, which supported the LRT and accumulation mechanisms. At 4-8 km, despite the



weak ozone destruction from OH radicals, the net chemical budget is negligible. This suggests a low photo-
chemical production, which is also supported by the negative $O_3$-CO correlation. According to the simulation
results and the $O_3$-CO correlation, a net contribution from STE can also influence the local ozone amount.
The two spring ozone interannual anomalies are believed to be influenced by the dynamical variations (SST
anomaly) during the El Niño events. The climate interacts with the distribution of tropospheric ozone through
temperature, humidity and dynamics.
**Acknowledgements**
We would like to thank the SCIAMACHY LNM $NO_2$, OMI/MLS, TES tropospheric ozone, IASI CO and teams
for providing the data. We acknowledge the two working staffs on models MACC reanalysis and MOZART-4.
Jia Jia acknowledges funding by CSC (China Scholarship Council) and scientific support from ESSReS (Earth
System Science Research School). We also acknowledge financial support provided by the University and State
of Bremen. We would like to thank Anne-Marlene Blechschmidt for helping preparing the wind field and
MACC data. Our gratitude goes to Prof. Christian von Savigny for giving comments during the preparation of
the manuscript.



**Appendix**

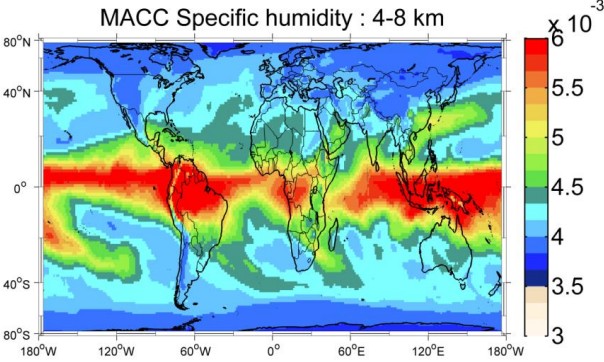

Figure A1. $O_3$ and CO partial column (4-8 km) time series at 21 N, 60 E over AS at April 2008 from MACC
reanalysis data. The plot is a 'point' example of Fig. 10.

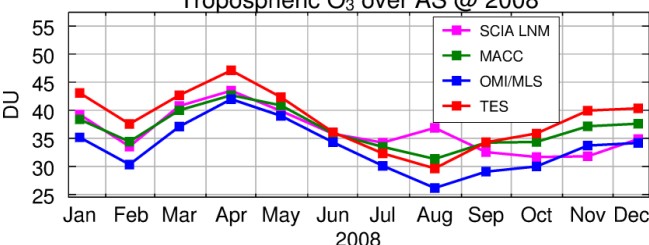

Figure A2. Same as Fig. 12 but for the year 2010.

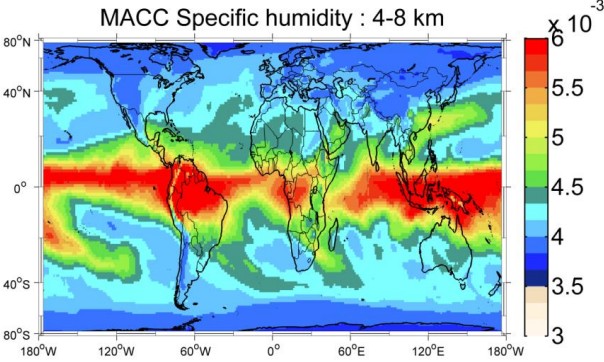

Figure A3. TOCs results over AS from SCIAMACHY Limb-Nadir-Matching, OMI/MLS, TES, and MACC
reanalysis data sets in 2008.



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
