# Peer review of "Tropospheric ozone maxima observed over the Arabian Sea during the pre-monsoon"

_Atmospheric Chemistry and Physics, 2016_

## Referee Comment (RC1) · Anonymous Referee #2 · 11 Nov 2016

General Comments:

The authors provide a detailed analysis for the variation of tropospheric ozone over Arabian Sea. They indicated that the maximum of spring time free tropospheric $O_3$ over AS is mainly driven by long-range transport, particularly, from India. I recommend the paper for publication after consideration of the points below.

Specific Comments:

1: Page 2, Line 10-12 It would be better to define the region of Arabian Sea in Figure 1. The enhancement is not very obvious. What is the reason for the 30x10 grid box line in Figure 1?

2: Page 2, Line 13-16 What is the relation between Southern Hemispheric biomass

burning with enhanced TOC over Arabian Sea? I assume detailed description and citations for Southern Hemispheric biomass burning are not necessary here.

3: Page 2, Line 20-21 If this phenomenon is not unique, but just a "well-known large scale phenomenon", why should we focus on it?

4: Section 2: Is there any evaluation study for the tropospheric O3 column provided by SCIAMACHY and OMI/MLS? The TOC is highly depended on the stratosphere-troposphere separation, which could be an issue for the unpolluted area.

5: Section 4 In Figure 3, can you add a panel to show the O3 variation over northern India? It seems that the O3 seasonality over AS is strongly correlated with the O3 seasonality over India (based on Figure 2). Instead of complex source analysis, I am wondering whether the variation of O3 over AS can be simply explained by the variation of O3 formation of India.

6: Section 4.3 The local chemistry production in lower troposphere is small (Figure 10), however, I am wondering whether the significant O3 production in upper troposphere has influences on lower tropospheric O3. Does lightning play a role in the O3 accumulation?

Technical comments:

1: Page 13, Line 11 Change Figure 6.11 to Figure 10

---

## Referee Comment (RC2) · Anonymous Referee #3 · 5 Jan 2017

This work describes variations in the tropospheric ozone column over the Arabian Sea using satellite data (SCIAMACHY, OMI/MLS, TES), MACC reanalysis data, MOZART-4 results & HYSPLIT trajectories and indicates that spring time ozone maximum over a region in the Arabian sea is largely (50% in 0-4 km and 20% in 4-8 km) due to LRT from India apart from some contribution from Middle East, Africa and Europe.

Major findings are based on MACC reanalysis data and MOZART-4 output (Hou et al. 2014). Quantitative contributions from different regions are mentioned, where detailed explanation and in-depth analysis for these regions was expected. It has been mentioned (page 2, line 20) that spring time maximum ozone is not unique but rather well-known. It has also been confirmed by other studies (line 32) that this is due to LRT from Middle East, Western India, Africa etc. Therefore, it was expected that this work should have provided some additional knowledge with more quantitative analysis.

[Figure]

There are significant differences in spatial distribution among SCIAMACHY, OMI/MLS and MACC reanalysis data over the Arabian Sea during spring. In my opinion, few of my below comments might be helpful in making it a better manuscript.

1) Introduction, Page 2, 2nd para – Enhancement in ozone over the Arabian Sea is not prominent. It would be better to include some discussion on enhancement in ozone over northern India. Additionally, higher ozone could also be seen over the Bay of Bengal in SCIAMACHY but not in OMI/MLS, any reason for it!

2) Section 3: CO and NO2 are ozone precursors and springtime CO column (Fig 2) values are observed to be much higher in the Bay of Bengal region than over the Arabian Sea. NO2 is also seen to be similar over AS and BoB. Any comment !!!. Additionally, CO is also higher in southern region of the Arabian Sea and southwest boundaries of India. But higher ozone (particularly SCIAMACHY) is seen in the northern part of the Arabian Sea and close to Oman, Yemen, Pakistan, etc. Why ozone is not higher close to western coast of India? Explanation should be added in this regard. Considering these facts, source of ozone maximum should be discussed.

3) Section 3: Figure 2 is for year 2008 and Figure 4 is for years 2006 and 2010. It will be good to add results for year 2008 in Figure 4 or make figure 2 for years 2006 or 2010. A significant difference in spatial distribution is seen over AS during spring/April. Figure 2 shows higher ozone (SCIAMACHY) having proximity to western region of AS, while this is not seen in Fig 4. It would be better to add monthly variations from MACC reanalysis (TOC, 0-4 km, 4-8 km, 8-12 km, 12-18 km) in a separate panel of figure 3.

4) Section 4.2: Line 10 ". . . . .CO and TOC are highly correlated . . . ." please clarify. This is in contradiction with statement at page 5 line 20 (". . . . CO and NO2 show a different . . . .").

Figure 6: Backward trajectories (mainly 2 and 4 km) are from southern India or some time from central region. Both these regions are showing much lower levels of ozone than those over northern India (Fig 1, 2, and 4).

Page 9, Line 14: Is this statement based on backward air trajectory of 15 km (Fig 6) alone or are there other supporting evidences? Lines 15-16 are also very qualitative.

Page 11, Line 7: Considering the broad region, it is better to name it as "South Asia". It would be beneficial to the reader to give some details on methodology adopted in estimating percentage contribution from different regions (Page 12). What/how was the background/reference levels considered while making these estimates?

5) Section 4.4: It would be worthwhile to list down the contribution from STE in the AS region. As mentioned that it is comparable with that of 'Euro_FT', which is significant (17%, 29%, 32%, and 31%).

Minor comments TPH information can be given in the caption of figure 4.

Referencing of figure needs to be in sequence. Figure 12 (@ page 10) is referred after figure 7 (@ page 9).

Hou et al. (2014) used MOZART-4 model run during 2000-2007, however this study used 1997-2007 model run. Please mention it explicitly if a separate run is made for this study.

Page 13: Change "Fig 6.11" to "Fig 10"

Conclusion: Line 12: Reference of Southern hemisphere biomass burning is appearing not to be relevant here. This has also been referred in the main text without much relevance.

---

## Author Comment (AC1) · 13 Mar 2017

The authors would like to thank Referee #2 for reviewing the manuscript.

Comments: The authors provide a detailed analysis for the variation of tropospheric ozone over Arabian Sea. They indicated that the maximum of spring time free tropospheric O3 over AS is mainly driven by long-range transport, particularly, from India. I recommend the paper for publica-tion after consideration of the points below.

Page 2, Line 10-12 It would be better to define the region of Arabian Sea in Figure 1. The enhancement is not very obvious. What is the reason for the 30x10 grid box line in Figure 1?

Response: The region of Arabian Sea was defined too far back (in Figure 2 as rectangle and in Page 7, Line 11). The definition is moved ahead to Page 2 Line 10-12. The sentence is now 'A tropospheric ozone maximum is observed over the Arabian Sea (AS, west side of the sub-continental India). ' The caption of Figure 1 is changed to '... and (right) OMI/MLS in 2008, with bold arrows pointing to AS. The AS region is defined as 10-20°N, 60-70°E in this study and is marked with red rectangle in Fig. 2). ' The 30x10 grid box line is a default setting of the software 'pglobal '.

Page 2, Line 13-16 What is the relation between Southern Hemispheric biomass burning with enhanced TOC over Arabian Sea? I assume detailed description and citations for Southern Hemispheric biomass burning are not necessary here.

Response: The study of AS tropospheric ozone columns is inspired by averaging TOCs over long time period, when the author found four main patterns in a global scale: plumes caused by biomass burning, by the anthropogenic pollution in midlatitudes northern hemisphere, by STE over Mediterranean, and the ozone enhancement over Arabian Sea. Plenty of studies and researches have helped us understanding the first three patterns of tropospheric ozone, while few were about the enhancement over Arabian Sea. Page 2, Line 12-18 states this motivation and aims to point out the outstanding magnitude of Arabian Sea ozone enhancement by com-paring it with the other three well-known patterns. The biomass burning pattern is one of them. The detailed citations may are not necessary, and several citations have been removed. The sentence is now '... and towards Australia (e.g., Fishman et al., 1986, 1991), 2) TOC attribut-ed to anthropogenic sources ...'

Page 2, Line 20-21 If this phenomenon is not unique, but just a "well-known largen scale phenomenon", why should we focus on it?

Response: The authors agree that this sentence weakened the importance of the AS ozone enhancement. The sentence is now 'Although the TOC enhancement over AS is an important global pattern of tropospheric ozone, the spring maxima in TOC are not unique over the AS representing rather a well-known large scale phenomenon in the

Northern Hemisphere. '

Section 2: Is there any evaluation study for the tropospheric O3 column provided by SCIA-MACHY and OMI/MLS? The TOC is highly depended on the stratosphere-troposphere sepa-ration, which could be an issue for the unpolluted area

Response: The evaluation studies for the tropospheric ozone column provided by SCIAMA-CHY and OMI/MLS were reported by Ebojie et al. (2014) and Ziemke et al. (2006) respec-tively. The monthly results for both data sets are within 5 DU comparing to ozonesonde measurements. The stratosphere-troposphere separation is operated by using tropopause height data. The tropopause height information for SCIAMACHY LNM and OMI/MLS TOC retrieval is derived from ECMWF and NCEP, respectively. The fact that different databases are used to retrieve tropopause height for SCIAMACHY and OMI/MLS is not expected to have a significant effect as tropical region has the most stable tropopause height compared to the other latitudes.

Section 4 In Figure 3, can you add a panel to show the O3 variation over northern India? It seems that the O3 seasonality over AS is strongly correlated with the O3 seasonality over India (based on Figure 2). Instead of complex source analysis, I am wondering whether the variation of O3 over AS can be simply explained by the variation of O3 formation of India.

Response: The TOC variations over northern India and AS from OMI/MLS are shown below (attached figure-1). Despite the similar seasonal pattern in general, the variation of ozone over AS cannot be simply explained by the variation of ozone formation of India. As discussed in the manuscript, the Northern India is a major source region for air masses over Bay of Bengal, but not for the Arabian Sea.

Section 4.3 The local chemistry production in lower troposphere is small (Figure 10), however, I am wondering whether the significant O3 production in upper troposphere has influences on lower tropospheric O3. Does lightning play a role in the O3 accumu-lation?

[Figure]

Response: The HYSPLIT trajectory forward model results (Fig. 7) indicate that the air mass-es in upper troposphere are transported towards the Pacific Ocean rather than sinking down-wards. In Page 14, Line 7-9, we also pointed out that ozone amount over ~11 km in pre-monsoon season has a zonal pattern. We conducted limited studies on lightning influence by checking the lightning flash rate data provided from LIS/OTD (Lightning Imaging Sensor/space borne Optical Transient Detector). The lightning over AS is negligible. Lightning can play a role by influencing the source region before long range transport. Barret et al. studied lightning NOx influence over Arabian Sea by using GEOS-Chem model. Ozone concentration at 565 hPa can decrease by 20 ppbv when turning off global lightning, while it remains almost unchanged without the local lightning. These results were presented in a poster and was not published in journals in accordance with our knowledge.

Technical comments: Page 13, Line 11 Change Figure 6.11 to Figure 10

Response: Typo corrected.

Besides of the comments that were given, the authors would like to clarify some con-tributors in the acknowledgements: '...We acknowledge the working staffs on MACC reanalysis, NCEP, and MOZART-4. ... We would like to thank Dr. Anne-Marlene Blechschmidt for her help. Our gratitude goes to Prof. Christian von Savigny for giving comments during the preparation of the manuscript. The authors acknowledge the North-German Supercomputing Alliance (HLRN) for providing HPC resources that have contributed to the research results reported in this paper'

Please also note the supplement to this comment:
http://www.atmos-chem-phys-discuss.net/acp-2016-786/acp-2016-786-AC1-supplement.zip

[Figure]

[Figure]

Fig. 1.

[Figure]

---

## Author Comment (AC2) · 13 Mar 2017

The authors would like to thank Referee #3 for reviewing the manuscript.

Comments: This work describes variations in the tropospheric ozone column over the Arabian Sea using satellite data (SCIAMACHY, OMI/MLS, TES), MACC reanalysis data, MOZART-4 results & HYSPLIT trajectories and indicates that spring time ozone maximum over a region in the Arabian sea is largely (50% in 0-4 km and 20% in 4-8 km) due to LRT from India apart from some contribution from Middle East, Africa and Europe. Major findings are based on MACC reanalysis data and MOZART-4 output (Hou et al. 2014). Quantitative contributions from different regions are mentioned, where detailed explanation and in-depth analysis for these regions was expected. It has been mentioned (page 2, line 20) that spring time maximum ozone is not unique but rather

well-known. It has also been confirmed by other studies (line 32) that this is due to LRT from Middle East, Western India, Africa, etc. Therefore, it was expected that this work should have provided some additional knowledge with more quantitative analysis. There are significant differences in spatial distribution among SCIAMACHY, OMI/MLS and MACC reanalysis data over the Arabian Sea during spring. In my opinion, few of my below comments might be helpful in making it a better manuscript.

Specific comments: Introduction, Page 2, 2nd para – Enhancement in ozone over the Arabian Sea is not prominent. It would be better to include some discussion on enhancement in ozone over northern India. Additionally, higher ozone could also be seen over the Bay of Bengal in SCIAMACHY but not in OMI/MLS, any reason for it!

Response: The enhancement in ozone over the Arabian Sea can be observed better in a 7-years averaged image as shown below (in attached figure-1). This enhancement is exciting as AS is a remote marine region. The enhancement in ozone over northern India is expected with anthropogenic pollution and biomass burning over Indo-Gangenic Plain. The authors agree that this en-hancement should also be mentioned. The sentence in Page 2 Line 12-18 is changed to '. . ., 2) TOC attributed to anthropogenic sources in the Northern Hemisphere, for instance, Northern India, and 3) the Mediterranean . . .' High ozone values can also be seen over the BoB in OMI/MLS during spring time (Figure 2). The higher values in SCIAMACHY seem to be contributed from summer-monsoon. General-ly, if any feature is seen by one instrument and not seen by another, the difference might be explained either by different samplings of the instruments or by quality problems in one of the data set. Here the second explanation is more probable. A validation with respect to inde-pendent data is needed to decide which data set is more correct. Unfortunately, no sufficient balloon/ship measurement was taken over BoB.

Section 3: CO and NO2 are ozone precursors and springtime CO column (Fig 2) values are observed to be much higher in the Bay of Bengal region than over the Arabian Sea. NO2 is also seen to be similar over AS and BoB. Any comment !!!. Additionally, CO

is also higher in southern region of the Arabian Sea and southwest boundaries of India. But higher ozone (particularly SCIAMACHY) is seen in the northern part of the Arabian Sea and close to Oman, Yemen, Pakistan, etc. Why ozone is not higher close to western coast of India? Explanation should be added in this regard. Considering these facts, source of ozone maximum should be discussed.

Response: One difference between CO and tropospheric ozone is that CO is emitted directly from pollutions, while ozone is produced through secondary photochemical reactions. CO is mostly observed near the boundary layer close to the emission sources; in this case, biomass burnings in Southeast Asia and Indo-Gangetic Plain. Srivastava et al. (2011) reported that transpo5rt from highly polluted Indo-Gangetic Plain is the major source for polluted air masses over Bay of Bengal. Thus, CO column values are much higher in the Bay of Bengal region. NO2, on the other hand, is a short lived trace gas, which is often observed at source region. The similar value of NO2 over AS and BoB is at its background level. At this situation, the present of higher CO in southern region of the Arabian Sea and southwest boundaries of India would not contribute to ozone production due to wet-deposition. The different spatial distribution among ozone precursors and ozone indicate a dynamic origin of high TOC over AS from long range transport of ozone. This sentence has been added to Page 5 Line 34. The air masses over AS is not mainly transported from northern India, but is transported from central/south India and Southwest Asia, i.e., Oman, Yemen, Pakistan, etc. Our study shows later that the source from India contributes to the lower altitude of ozone over AS, while Southwest Asia and Europe are the major sources for ozone columns from 4 km above (Fig. 9). Therefore, ozone over AS is higher close to Southwest Asia instead of western coast of India. The authors agree that this ozone distribution is worth to be pointed out and explained in the manuscript. We have added 'The spring TOC are higher over western AS than over eastern AS (Fig. 2). This distribution is further discussed in Sect. 4.2.' in Page 5, Line 19 and '. . . with over 20% contribution. Since 'Euro_FT' is the major source of TOC over AS, higher TOCs are observed over the western AS that is close to Southwest Asia, i.e., Oman, Yemen, Pakistan, etc., instead

of the eastern AS near west coast of India. . . .' in Page 12, Line 13.

Section 3: Figure 2 is for year 2008 and Figure 4 is for years 2006 and 2010. It will be good to add results for year 2008 in Figure 4 or make figure 2 for years 2006 or 2010. A significant difference in spatial distribution is seen over AS during spring/April. Figure 2 shows higher ozone (SCIAMACHY) having proximity to western region of AS, while this is not seen in Fig 4. It would be better to add monthly variations from MACC reanalysis (TOC, 0-4 km, 4-8 km, 8-12 km, 12-18 km) in a separate panel of figure 3.

Response: The results for year 2008 are added in Figure 4. Both satellite and model shows maxima in April (Fig. A3). As model data preparing is time consuming, we have chosen April instead of long time series. Analyzing monthly variations from MACC reanalysis data cannot help better understanding the spatial distribution differ-ences seen between Fig.2 and Fig. 4 since the time series represent a regional average. Never-theless, the authors found this review very interesting, we will consider putting this idea into the framework of the next study.

Section 4.2: Line 10 ". . .CO and TOC are highly correlated. . ." please clarify. This is in con-tradiction with statement at page 5 line 20 (". . .CO and $NO_2$ show a different. . .").

Response: The sentence '. . .CO and TOC are highly correlated. . .' is deleted.

Figure 6: Backward trajectories (mainly 2 and 4 km) are from southern India or some time from central region. Both these regions are showing much lower levels of ozone than those over northern India (Fig 1, 2, and 4).

Response: The high ozone over northern India (mainly over Indo-Gangetic Plain) is mainly due to the anthropogenic pollution with intensive population and rapidly growing industry. It is the most polluted region not only for ozone but also other trace gases. Despite the TOC values over northern India are much higher than in other surrounding regions, our study shows that the source of increased TOC in AS region are located in central/southern India.

Page 9, Line 14: Is this statement based on backward air trajectory of 15 km (Fig 6) alone or are there other supporting evidences? Lines 15-16 are also very qualitative.

Response: Thank you very much for this comment. The conclusions are based on the trajec-tory of AS in different years and at different altitudes in April. Figure 6 and 7 only give ex-amples of the trajectories. The statement at Page 9 Line 14 is indeed misleading the reader. The sentence has been changed into 'The higher tropospheric ozone (12-18 km) is found in air which was uplifted and transported not only from the North Africa, but also from the North Indian Ocean and Southeast Asia.'

Page 11, Line 7: Considering the broad region, it is better to name it as "South Asia". It would be beneficial to the reader to give some details on methodology adopted in estimating percentage contribution from different regions (Page 12). What/how was the back-ground/reference levels considered while making these estimates?

Response: We agree that the name 'South Asia' would be more appropriate. Unfortu-nately, due to technical reasons it is extremely time expensive to replace the name in the plot. As this is not crucial for the general content of the paper, we decided to keep the name 'India'. We have added a sentence in Page 11, Line 11 to clarify: 'The se-lected names are rather symbolic and do not pretend to be geographically fully correct.' The methodology in estimating contributions from different regions is briefly mentioned in page 4, line 34. We added a sentence in page 12 Line 6 to help readers understand more about the percentage estimation. 'The percentage contributions from different regions are calculated by first estimating the ozone concentration (in ppbv) from a sep-arated region using tagged tracer method (Sudo and Akimoto, 2007), then dividing this value by the sum of ozone concentrations. '

Section 4.4: It would be worthwhile to list down the contribution from STE in the AS region. As mentioned that it is comparable with that of 'Euro_FT', which is significant (17%, 29%, 32%, and 31%). .

Response: The contributions from STE are now listed as you can see here: This

information is added on Page 16, line 11. '. . . is comparable with the ones transported from 'Euro_FT' in most altitude ranges (STE to Euro_FT contributions are 0.955 at 0-4 km, 0.861 at 4-8 km, 0.997 at 8-12 km, and 16.858 at 12-18 km). The STE origin . . .'.

Minor comments: TPH information can be given in the caption of figure 4.

Response: The TPH information has been added in the caption of figure 4 as: 'Ozone partial columns (TOC: 0 km - TPH($\sim$17 km over AS) column of tropospheric ozone in the left panels and 0-8 km . . .'

Referencing of figure needs to be in sequence. Figure 12 (@ page 10) is referred after figure 7 (@ page 9).

Response: The text at Page 10, line 5 has been changed to '... by Hydroxyl radicals (OH). This is further . . .', where figure 12 is no longer referred before figure 7.

Hou et al. (2014) used MOZART-4 model run during 2000-2007, however this study used 1997-2007 model run. Please mention it explicitly if a separate run is made for this study.

Response: The data used in this study was obtained by Zhu et al.(2016) using the same sce-nario as Hou et al. (2014). This run was not made specifically for this study. The description is added in Page 4, Line 30-37: '. . . from the surface to approxi-mately 2 hPa. The data used in this study are simulation results obtained by Zhu et al. (2016), using MOZART-4 model for 1997-2007. The chemical initial condition in 2000 and emissions from 1997 to 2007 used . . . it's transport, chemical transformation and surface deposition. More details of the simulation settings can be found in Hou et al. (2014).'

Page 13: Change "Fig 6.11" to "Fig 10"

Response: Typo has been corrected

Conclusion: Line 12: Reference of Southern hemisphere biomass burning is appearing

not to be relevant here. This has also been referred in the main text without much relevance.

Response: The study of AS tropospheric ozone columns is inspired by averaging TOCs over long time period, when the author found four main patterns in a global scale: plumes caused by biomass burning, by the anthropogenic pollution in midlatitudes northern hemisphere, by STE over Mediterranean, and the ozone enhancement over Arabian Sea. Plenty of studies and researches have helped us understanding the first three patterns of tropospheric ozone, while few were about the enhancement over Arabian Sea. Page 2, Line 12-18 states this motivation and aims to point out the outstanding magnitude of Arabian Sea ozone enhancement by com-paring it with the other three well-known patterns. The biomass burning pattern is one of them.

Besides of the comments that were given, the authors would like to clarify some contributors in the acknowledgements: '...We acknowledge the working staffs on MACC reanalysis, NCEP, and MOZART-4. ... We would like to thank Dr. Anne-Marlene Blechschmidt for her help. Our gratitude goes to Prof. Christian von Savigny for giving comments during the preparation of the manuscript. The authors acknowledge the North-German Supercomputing Alliance (HLRN) for providing HPC resources that have contributed to the research results reported in this paper'

Please also note the supplement to this comment:
http://www.atmos-chem-phys-discuss.net/acp-2016-786/acp-2016-786-AC2-supplement.zip

[Figure]

[Figure]

**Fig. 1.** 7 years average for TOC retrieved from (left) SCIAMACHY LNM and (right) OMI/MLS.